# Endogenous Double-Stranded RNA

**DOI:** 10.3390/ncrna7010015

**Published:** 2021-02-19

**Authors:** Shaymaa Sadeq, Surar Al-Hashimi, Carmen M. Cusack, Andreas Werner

**Affiliations:** Biosciences Institute, Medical School, Newcastle University, Newcastle upon Tyne NE2 4HH, UK; s.k.sadeq2@newcastle.ac.uk (S.S.); s.o.t.al-hashimi2@newcastle.ac.uk (S.A.-H.); Carmenmariacusack@gmail.com (C.M.C.)

**Keywords:** double-stranded RNA (dsRNA), innate immunity, repetitive DNA elements (RE), antisense transcript

## Abstract

The birth of long non-coding RNAs (lncRNAs) is closely associated with the presence and activation of repetitive elements in the genome. The transcription of endogenous retroviruses as well as long and short interspersed elements is not only essential for evolving lncRNAs but is also a significant source of double-stranded RNA (dsRNA). From an lncRNA-centric point of view, the latter is a minor source of bother in the context of the entire cell; however, dsRNA is an essential threat. A viral infection is associated with cytoplasmic dsRNA, and endogenous RNA hybrids only differ from viral dsRNA by the 5′ cap structure. Hence, a multi-layered defense network is in place to protect cells from viral infections but tolerates endogenous dsRNA structures. A first line of defense is established with compartmentalization; whereas endogenous dsRNA is found predominantly confined to the nucleus and the mitochondria, exogenous dsRNA reaches the cytoplasm. Here, various sensor proteins recognize features of dsRNA including the 5′ phosphate group of viral RNAs or hybrids with a particular length but not specific nucleotide sequences. The sensors trigger cellular stress pathways and innate immunity via interferon signaling but also induce apoptosis via caspase activation. Because of its central role in viral recognition and immune activation, dsRNA sensing is implicated in autoimmune diseases and used to treat cancer.

## 1. Introduction

If an endeavor has “Buckley’s chance”, no one in Melbourne would bet any money on it, as the odds to succeed are close to zero. The phrase “Buckley’s chance” refers to William Buckley, an English convict who was deported to Australia. He escaped and lived with an Aboriginal tribe for more than 30 years. The chances of survival were, indeed, very slim for Buckley from the start; he was pursued and shot at when he escaped, and then he had to survive in the scorching Australian summer with little water and no food. Finally, he had to learn to communicate with the Aboriginal people and win their respect. In many ways, the unlikely survival story of William Buckley could stand as a metaphor for the development of spurious transcripts into “functional” long non-coding RNAs (lncRNAs) in a treacherous cellular environment.

The genome of complex organisms is riddled with repetitive sequences related to endogenous retroviruses (ERVs) and DNA transposons. They constitute a large part of the genome; in humans, 50–70% are repetitive or repeat-derived [1,2] and are largely responsible for the variation in genome size of complex organisms [3,4]. Despite the fact that the two classes of transposable elements (ERV and DNA transposons) can be grouped into superfamilies that are present in all taxa and then further into families and subfamilies, particular variants of transposable elements are species-specific.

The vast majority of transposons and retroviruses are inactivated through truncations and point mutations. In humans, only about 100 L1 retrotransposons (of about 500,000) are full-length, and less than 10 retained retro-transposition potential [5,6]. Hence, the repetitive, low-complexity part of the genome is often referred to as “junk DNA” [7].

Whether the vast graveyard of transposable elements actually represents “junk”, functional elements or recyclable material constitutes an ongoing scientific debate [8,9]. Two important observations, however, are uncontested and particularly relevant in the context of long non-coding RNAs. First, the insertion of an ERV into the host genome affects transcriptional activity around the insertion site, thus creating the pressure to mitigate the overwhelmingly deleterious consequences of the interference [10]. Second, the remnants of transposable elements contain regulatory sequences such as weak promoters and enhancers or polyadenylation sites, and thus, a large proportion of the repetitive genome is being transcribed at a very low level [11,12]. In a sense, pervasive transcription may create opportunities to salvage genetic material in the form of long non-coding RNAs [13]. Accordingly, 75% of mature human lncRNA sequences contain an exon originating from transposable elements (TEs) [14,15]. Comparatively, the percentage of transcripts with TE material in 5′ and 3′ untranslated regions (UTRs) is substantially lower, with 8.44% in the 5′ UTR and 26.74% in the 3′ UTR [15]. The vast majority of the transcripts are quickly degraded because they lack protective modifications such as splicing, polyadenylation and capping that would also license them for export from the nucleus. Because of repetitive sequence content as well as bi-directional transcription, the spurious transcripts are prone to form both intra- and intermolecular double-stranded RNA (dsRNA) structures. Alternatively, the association with local cellular components such as chromatin remodeling complexes [16,17] may increase the stability and chances to escape degradation [18]. This brief review discusses the former outcome of pervasive transcription, the formation of endogenous dsRNA, which may trigger a cellular antiviral response; the focus will be on observations in humans and mice. It aims to draw a bigger picture rather than drilling into details.

## 2. Sources of Endogenous dsRNA

The detection and quantification of dsRNA requires specific tools such as specific antibodies or dsRNA-binding proteins [19,20]. After immune purification, RNA can be analyzed by high-throughput sequencing or conventional methods such as cloning or RT-PCR. An alternative strategy to investigate nuclear dsRNA uses adenosine-to-inosine (A-to-I) editing to identify double strand formation [21,22]. Single- or double-strand specific RNases in combination with RT-qPCR provide an additional tool to demonstrate RNA hybrids. Unfortunately, RNA purification prior to nuclease treatment introduces a positive or negative bias for dsRNA (depending on the specific methodology), making quantitative assays difficult to interpret [23].

There are three main sources of endogenous dsRNA: mitochondrial transcripts, repetitive nuclear sequences, including short and long interspersed elements (SINEs, LINEs), and endogenous retroviruses (ERVs) as well as natural sense–antisense transcript pairs.

### 2.1. Mitochondrial Transcripts

Human mitochondria have a circular genome of 16,566 bp, with a guanine-rich heavy strand and a guanine-poor light strand, depending on buoyant density. Both strands are equally transcribed, resulting in complimentary transcripts that may bind to each other, though the light strand undergoes rapid degradation. Complementarity encompasses the length of the entire mitochondrial genome, as shown by electron microscopic analysis [24,25]. The mitochondrial DNA encodes 13 genes, 12 of which are encoded by the heavy strand and one by the light strand [26]. Under physiological circumstances, the light strand is rapidly degraded by two enzymes, polynucleotide phosphorylase (PNPase) and the helicase HSuv3 [27]. PNPase is located in the inter-mitochondrial membrane space, thus being well-placed to play an important role in preventing the escape of dsRNA into the cytoplasm. Mitochondrial RNA is a potent stimulator of the innate immune system, especially in dendritic cells and Toll-like receptor (TLR)-expressing cells [28] via a protein kinase R (PKR)-modulated interferon response. Conversely, inhibition of HSuv3 resulted in an increase in dsRNA without triggering an interferon response, which suggests that the increased levels of dsRNA remained sequestered within the mitochondria [19]. These findings are underpinned by the knockout of PNPase or Suv3 that leads to an accumulation of dsRNA in the cytoplasm and an altered immune response [29]. Moreover, patients with bi-allelic PNPase variants showed increased levels of unprocessed mitochondrial transcripts and an enhanced expression of interferon-stimulated genes [30].

Mitochondrial dsRNA formation was also demonstrated using fCLIP-seq, an approach which entails formaldehyde cross-linking of PKR-bound dsRNA followed by high-throughput sequencing. Most of the dsRNA bound to PKR mapped to the mitochondrial genome. The mitochondrial origin of the RNA was corroborated by the lack of A-to-I edited nucleotides, as mitochondrial dsRNA is not subjected to adenosine deaminase acting on RNA (ADAR)-dependent editing [31]. Collectively, these findings established that mitochondria are an important source of dsRNA which may be released into the cytoplasm upon stress-mediated mitochondrial permeabilization [32].

### 2.2. Repetitive DNA Sequences

For dsRNA originating from nuclear DNA, A-to-I editing provides an accurate readout to assess genome-wide dsRNA formation [33]. In humans, 62.9% of all edited sites map to repeat regions, including SINEs, LINEs, endogenous retroviruses and DNA transposons, whereas protein coding transcripts are hardly edited at all (Figure 1). Overall, editing shows distinct species’ variability and depends on the nature of the repetitive elements rather than the complexity of the organism [33,34].

SINEs: The most common sources of dsRNA in human cells are Alu repeats, the most abundant class of short interspersed nuclear elements [35] (Figure 1). Alu elements are approximately 300 nucleotides in length and contain two 7SL RNA genes including short A-rich stretches [36,37].

Alu repeats are commonly found in intergenic regions (autonomous) as well as in introns and UTRs of genes (mRNA-embedded elements) [38]. Autonomous Alu elements constitute a small portion of the repetitive genome and are highly induced by viral infection, heat shock and cycloheximide treatment [39]. Stress enhances the activity of the RNA polymerase lll (viral infection) or increases the chromatin accessibility of Alu elements (heat shock), which is reversed with recovery from stress [40]. As compared to autonomous Alus, embedded Alu elements represent a higher proportion of repeated sequences. Because of their enrichment in UTRs, embedded Alus play an important function in gene expression via the stabilization of mRNA, as well as its localization and translation [38,41].

The repetitive nature of Alu insertions allows the formation of predominantly intramolecular dsRNA, which is recognized by the nuclear isoform of ADAR [42,43]. In addition, PKR-fCLIP sequencing showed that more than 20% of dsRNAs associated with PKR derive from Alu repeats [31]. The Alu-dependent dsRNAs are not long enough to trigger efficient oligomerization and activation of melanoma differentiation-associated gene 5 (MDA5). In contrast, a mutated form of MDA5 that shows greater tolerance towards mismatches in the RNA hybrid has been linked to immune hypersensitivity and autoimmune disease (Aicardi–Goutières syndrome, [44]).

ERVs: Human endogenous retroviruses share a comparable structure with exogenous retroviruses, the protein coding genes gag, pro (protease), pol and env flanked by two terminal repeats (5′ and 3′ LTR) (Figure 1). ERVs comprise up to 8% of the human genome; however, most open reading frames (ORFs) are mutated [45]. Nevertheless, ERV-related transcripts can be detected in most human tissues [46], particularly when repressive DNA methylation is inhibited. In contrast to the mutated protein coding genes, ERV-related LTRs have retained their promoter activity and provide alternative transcriptional control elements for cellular genes or drive the production of non-coding cellular RNA [45,47].

LTR promoters are bi-directional and can lead to widespread dsRNA formation [48,49]; alternatively, two adjacent ERVs in opposite orientations could fold back and form a hairpin structure [31]. Although ERVs are not a very common source of dsRNA, the activation of LTR promoters and subsequent dsRNA formation still have significant clinical consequences. For example, transcription of ERVs can be triggered by DNA methyl transferase inhibitors such as Azacitdine and Decitabine through demethylation and activation of ERV promotors [50]. Induction of ERV expression results in activation of the mitochondrial antiviral signaling protein/interferon regulatory factors (MAVS-IRFs) pathway via MDA5 and, to lesser extent, retinoic acid-inducible gene I (RIG1). This “viral mimicry” is exploited for the treatment of many cancers such as melanoma and colorectal carcinoma by activating an innate immune response against cancer cells [51].

LINEs: Long interspersed nuclear elements (LINEs) are 6–7 kb in size and constitute up to 20% of the human genome. Full-length copies contain two open reading frames (ORF1 and ORF2) which encode proteins essential for retro-transposition [52] (Figure 1). ORF1 makes a 40-kDa RNA-binding protein (RBP 40) which plays an important role in activating the host innate immune system, while ORF2 encodes an endonuclease and the reverse transcriptase [53]. Transcription is driven by a promoter that harbors several transcription factor binding sites as well as a CpG island. Most LINEs are inactive because of truncations, mutations and rearranged copies; however, a small number of elements are functional [54].

The exact mechanisms by which LINEs form a double-strand configuration is unknown; some studies hypothesize that they form hairpin structures when two complementary LINEs are present in the same transcript. Alternatively, two LINEs on two different transcripts close to each other can hybridize [55]. This idea is supported by fCLIP sequencing data showing that the distance between two LINEs interacting with PKR is much shorter than the space between random copies [31]. Furthermore, LINE elements have the ability to fold back on their 5′ region, forming stable hairpin structures that are recognized by PKR [20].

LINEs associate with various dsRNA binding proteins, mostly PKR and MDA5, and their expression has been linked to the activation of an interferon 1 response [31]. Moreover, extensive editing of LINEs by ADAR has been shown using ADAR–CLIP sequencing [56,57]. Although LINEs only give rise to 3% of cellular dsRNA as compared to 67% from SINEs, they are linked to many human diseases [44].

Natural antisense transcripts: According to the gencode biotype definition, antisense transcripts are “transcripts that overlap the genomic span (i.e., exon or introns) of a protein-coding locus on the opposite strand”. This definition excludes protein-coding antisense transcripts and read-through transcripts from tail-to-tail arranged gene pairs; if those are included, 40–70% of loci show bi-directional transcription [58,59]. Hence, if a sense/antisense transcript pair is co-expressed in the same cell, dsRNA structures are potentially formed (Figure 2). To what extend hybridization actually occurs is controversial and rather challenging to demonstrate experimentally.

Before the dawn of the genomics era, natural antisense transcripts were studied in the context of parental imprinting. Early ground-breaking work demonstrated that the expression of the antisense transcript was associated with the silencing of the related sense transcript on the same allele. Experimental silencing of the antisense transcript (Airn, Kcnq1ot1, for example) abolished parental imprinting and led to bi-allelic expression of the entire cluster, not only of the complementary gene [60,61]. Similar observations were made with non-imprinted genes; a deletion in the genome of a patient with α-thalassemia placed the constitutively active LUC7L (Putative RNA-Binding Protein Luc7-Like) directly downstream of the HBA2 (Hemoglobin 2A) gene. The ectopic expression of LUC7L produced an antisense transcript complementary to HBA2, causing hypermethylation of the CpG-rich promoter and transcriptional silencing of the gene [62]. Likewise, the promotor of the tumor suppressor gene p15 (CDKN2B, Cyclin Dependent Kinase Inhibitor 2B) is hypermethylated and silenced in various tumors, associated with the expression of the antisense transcript p15-AS (CDKN2B-AS1) [63]. Silencing was found to be independent of Dicer, and the fact that the entire CDKN2B gene is imbedded in an intron of CDKN2B-AS1 argues against a role of dsRNA formation in an antisense transcript-mediated regulatory mechanism [63,64].

On the other hand, there is increasing evidence of dsRNA formation as the result of antisense transcription from both genomics studies and examples of specific sense–antisense transcript pairs. Early studies on the genome-wide expression of natural antisense transcripts followed a strategy where complementary full-length transcripts and expressed sequence tags in whole-data repository searches were identified [65,66]. The formation of dsRNA is inferred by the observation that natural antisense transcripts are significantly under-represented on the X chromosome of both humans and mice, whereas no such bias was found for sense–antisense pairs that lacked exonic complementarity [65,66]. Accordingly, dsRNA formation between processed transcripts represents a feature with a positive (accumulation on autosomes) or negative impact (reduction on X chromosomes) on evolutionary selection. The implications of dsRNA formation in the context of antisense transcription have been discussed including RNA masking, RNA editing, RNA interference as well as the stimulation of an innate immune response [67]. RNA masking is generally associated with a concordant expression of sense and antisense transcripts, often by interfering with the inhibitory action of miRNAs [68,69]. The latter three mechanisms (RNA editing, RNA interference and immune response) induce a discordant expression of the sense–antisense transcript pair (“antisense inhibits sense”) (Figure 2).

There is a steadily increasing number of reports on specific sense–antisense pairs where dsRNA formation is implicated in a regulatory interaction between the two transcripts. In line with the proposed mechanisms, both concordant and discordant expression of the complementary transcripts have been observed [59]. An example of a stimulatory interaction described in detail is the interplay between the transcript for β secretase-1 (BACE1) and its natural antisense transcript (BACE1-AS) in the context of Alzheimer’s disease pathophysiology. The antisense transcript protects BACE1 mRNA from miR-485-5p-induced degradation, and because of the increased β secretase, more β amyloid 1-42 was produced. In line with the mechanism, the levels of BACE1-AS were elevated in patients with Alzheimer’s disease [70,71]. Other selected examples of antisense transcripts masking miRNA binding sites are listed in Piatek et al. [72]. Natural antisense transcripts can also stabilize the sense transcript by blocking the binding of RNA decay-promoting factors [73]. This mode of action is exemplified by the interaction between the tumor suppressor gene PDCD4 (Programmed Cell Death 4) and its antisense transcript PDCD4-AS1 in mammary epithelial cells. The antisense transcript blocks the binding of human antigen R (HuR), which, in turn, stabilizes the sense mRNA and leads to increased PDCD4 expression [73]. Accordingly, PDCD4-AS1 expression is decreased in breast cancer patients and is low in mammary epithelial cells.

The mechanisms that lead to the degradation of the sense transcript generate specific products that can be experimentally assessed at a large scale, i.e., A-to-I conversions for editing, short RNAs for RNA interference and sequencing of RNA bound to protein kinase R. However, only limited evidence supports that these mechanisms are involved in processing RNA hybrids between genic sense and antisense transcripts—at least in a specific experimental context [74,75]. There are a few examples where the involvement of Dicer or ADAR has been experimentally tested for specific bi-directionally transcribed loci including the gene pairs glutaminase (GLS)/GLS-AS or sodium/phosphate co-transporter and a read-through transcript from profilin 3 (Slc34a1/Pfn3) [74,76]. Low levels of GLS-AS and enhanced expression of GLS in patients with pancreatic cancer predict a poor clinical outcome. The underlying mechanism was investigated in PANC-1 cells (human pancreatic cancer cell line-1). Accordingly, dsRNA formation occurs in the nucleus and both ADAR and Dicer can process the hybrid, resulting in a decrease in GLS sense mRNA and encoded glutaminase. Enhanced levels of glutaminase are observed under nutrient stress and related to tumorigenesis [74]. With regard to the Slc34a1/Pfn3 locus, there is little evidence that the antisense transcript is involved in the physiological regulation of the Na-phosphate cotransporter. Depending on the model system, both RNA interference and transcriptional interference can be observed. The fact that both transcripts are lowly expressed in testis may indicate that the sense–antisense interaction is biologically relevant in male germ cells, where the vast majority of natural antisense transcripts are expressed [76].

Despite the ever-increasing number of mechanistically established sense–antisense interactions, there is still a huge gap between the number of characterized examples and the thousands of sense–antisense gene pairs. An interesting set of articles have recently revived the idea that natural antisense transcripts and the potential dsRNA formation feed into a common mechanism(s) that merits selection, as seen with the X-chromosome bias or—more generally—the weak evolutionary conservation of sense–antisense arrangements [77].

Work in a preprint by S Pillay investigated the role of natural antisense transcript expression during early zebrafish embryogenesis and divided the RNAs into two groups with negative and positive correlation with sense transcript abundance, respectively [78]. Positively correlated transcripts are predominantly associated with house-keeping genes, whereas the transcripts with discordant expression are maternally expressed and are complementary to developmental genes. Based on the finding that the discordantly regulated transcripts were enriched in the cytosol, the authors speculate that these natural antisense transcripts act in a similar way as miRNAs to silence ectopic expression of developmental genes [78]. Another study in our own lab focused on dsRNA formation in mouse testis and involved enrichment of dsRNA using the J2 antibody followed by deep sequencing. We found that dsRNA was predominately present in pachytene spermatocytes and that the dsRNA transcriptome in testis was fundamentally different from the one in somatic liver cells. In both cell types, dsRNA was derived from mitochondrial transcription, though in testis, mRNA-related signals were clearly more abundant than in liver. Moreover, we could establish an association between dsRNA, antisense genes and endogenous siRNAs (small interfering RNAs)—again, the link was weaker or insignificant in liver cells (Werner et al., under revision). Importantly, both investigations focused on native tissues and cells, developing male germ cells and early zebrafish embryos, respectively. Both systems display low levels of DNA methylation [79,80] and transcriptional activity that is distinct from “normal” somatic cells. Moreover, testis male germ cells are immune privileged and tolerate dsRNA without activating innate immunity [81]. It is intriguing to speculate that natural antisense transcripts and dsRNA formation play a role in mitigating the consequences of the genome-wide transcriptional changes. Findings in zebrafish and mouse testis also suggest that dsRNA may have a fundamentally different impact in somatic cells.

The different handling of dsRNA in germ cells versus somatic cells has been experimentally corroborated using transgenic mice expressing a construct with a long hairpin the 3′ UTR. In mouse oocytes, dsRNA was processed into siRNAs, whereas in somatic cells, a small fraction was A-to-I edited. An interferon (IFN) response was only observed after high-level expression of the hairpin construct in a transfected human cell line (HEK293) [82]. A germ cell-specific biological role of dsRNA and endo-siRNAs is also supported by low siRNA sequencing of both female and male germ cells [83,84].

## 3. Proteins Binding dsRNA

The structure of dsRNA adopts an A-form duplex with a narrow major groove (4-Å width) and wide minor groove (10–11-Å width). As a consequence, dsRNA-binding proteins are generally unable to form base pair-specific interactions and recognize the backbone rather than sequence motives [85]. However, a few examples such as ADAR2 or STAUFEN recognize specific base pairs in the minor grove of the duplex [86,87]. Moreover, additional structures such as the Cap or RNA base modifications affect the binding of dsRNA-binding proteins and help the distinction between viral RNA and endogenous RNA hybrids. The dsRNA-binding protein families include RIG-I-like receptors (RLRs), PKR, ADAR, oligo adenylate synthetase (OAS), Dicer, Drosha and other helicases [20]. We focus here on the dsRNA-binding proteins that create a link between pathogenic dsRNA formation and the immune system (Figure 3). As part of the host defense against invading pathogens, these dsRNA sensors are also linked to a number of inflammatory and autoimmune diseases [88,89].

RIG-I-like receptors (RLRs): The protein family of retinoic acid-inducible gene-like receptors (RLRs), also called cytosolic RNA sensors, includes RIG1, MDA5 and laboratory of genetics and physiology 2 (LGP2). The latter lacks two caspase recruitment domains which are essential for downstream signaling. Consequently, LGP2 plays a regulatory role rather than an effector function in a dsRNA response [90]. The two main sensors that trigger a dsRNA inflammatory response are RIG1 and MDA5, which will be briefly introduced here [91].

RIG1 and MDA5 are members of the DExD/H box helicase family and contain five specific protein domains: from the N terminus, two caspase recruitment domains (CARDs), which participate in antiviral signaling, a DEAD-like helicase superfamily ATP-binding domain (DExDc), a helicase domain (HELICc) and a zinc-binding C-terminal domain [92]. In the non-signaling state, the two N-terminal domains are auto-repressed and unable to bind to mitochondrial antiviral signaling (MAVS) protein, a protein involved in the cellular innate antiviral defense. The auto-repression is abolished by the release of the N-terminal domains upon binding to dsRNA via helicase and the C-terminal domains [93].

RIG1 and MDA5 share the same signaling pathway but identify a discriminate group of dsRNA. Dimerization of RIG1 only takes a 300-base-pair duplex but requires a 5′ triphosphate group at the RNA end [94]. The triphosphate group is normally found in RNAs but is 7-methyl guanosine-capped in most eukaryotic mRNAs in the cytosol. Viral RNA usually lacks this modification. Recognition of dsRNA by MDA5 does not depend on the triphosphate group but requires a longer stretch of dsRNA (500–1000 bp) for a process of nucleation and filament assembly to be activated [95].

RIG1 and MDA5 activation leads to oligomerization of CARD domains, which, in turn, produces a platform for the generation of MAVS filaments at the mitochondrial membrane [96]. This triggers two main cascades, one activating nuclear factor κB and the transcription of proinflammatory genes, the other leads to the phosphorylation of interferon regulatory factors 3/7 (IRF3/7) and the stimulation of interferon gene expression [97].

Protein kinase regulated by RNA: PKR, also referred to as eukaryotic translation initiation factor 2-alpha kinase 2, EIF2AK2, is activated by binding to dsRNA and its gene expression is induced by interferon [98]. PKR includes two N-terminal RNA-binding motifs (RI and RII) and a catalytic kinase domain at the C-terminus [99]. The dsRNA-binding domains can interact with adjacent minor grooves of dsRNA by binding to the phosphate and ribose backbone independent of the base sequence [100].

Activation of the enzymatic activity of PKR requires an RNA duplex of at least 33 bp. Activation efficiency increases up to 85 bp and decreases with longer duplexes or high concentrations of dsRNA because of a dilution effect that reduces the chances of PKR dimerization [101]. PKR recognizes all types of dsRNA, but the majority of PKR was bound to dsRNA of mitochondrial origin, followed by IRAlus (inverted- repeat Alu elements, 20%) [31].

Binding of PKR to dsRNA induces a conformational change which displaces the inhibitory dsRNA binding domain from the catalytic kinase domain [102]. Moreover, homodimerization results in auto-phosphorylation and activation of PKR. The activated kinase dissociates from dsRNA and phosphorylates eukaryotic translation initiation factor 2A (EIF2A) at serine 51 and triggers global translational shut-down [103]. Alternatively, PKR phosphorylation may lead to Fas-associated via death domain (FADD)/caspase 8-mediated activation of caspases 3/7 and, ultimately, apoptosis [104,105].

Adenosine deaminase acting on RNA (ADAR): Members of the ADAR protein family catalyze the conversion of A to I in dsRNA. In humans, there are three ADAR genes: ADAR1, ADAR2 and ADAR3, with ADAR1 being interferon-inducible [106,107]. All of the three ADAR proteins contain two or three dsRNA-binding domains and a C-terminal deaminase domain. Moreover, ADAR1 has one or two N-terminal Z-DNA-binding domains and ADAR3 contains an arginine-rich region [108].

Transcription of ADAR is driven by interferon inducible- and constitutively active promoters [109]. ADAR1 is ubiquitously expressed in human tissues and predominantly targets dsRNA formed by IRAlus in the 3′ UTR of the mRNAs. Around 97.7% of editing occurs in non-protein-coding regions [110,111]. ADAR2 expression is highest in the brain and is directly linked to site-specific base changes of neurotransmitter receptor transcripts with functional and phenotypic consequences [112,113]. Additional targets have been identified in the brain and other tissues, but the consequences of editing are less well established [114]. ADAR2 accounts for 25% of the editing in non-repetitive sites in protein-coding transcripts [111]. ADAR3 is exclusively expressed in the brain; the enzyme lacks catalytic activity and its main role appears to be the inhibition of ADAR2 by competition for dsRNA binding [115].

ADAR antagonizes apoptosis by counter-balancing the activation dsRNA sensors and the stimulation of inflammatory and apoptotic signaling [116]. In a negative feedback mechanism, interferon stimulates ADAR that binds to and melts dsRNA, thus competing with other dsRNA sensors [117]. Despite compartmentalization of dsRNA and the various other strategies to distinguish intrinsic dsRNA from viral insurgents, there are still various pathologies with an underlying inflammatory phenotype potentially linked to endogenous dsRNA. Two examples where dsRNA plays a role in disease development but also offers treatment avenues are cancer and autoimmune diseases.

## 4. Physiological and Pathophysiological Roles of dsRNA

Apart from stimulating an antiviral response, there is growing evidence to suggest that dsRNA contributes to physiological cell growth and function, depending on the length, abundance and location of dsRNA within the cell [118,119]. In this context, the activation of PKR and downstream interferon signaling as well as TLR3 activation by cytoplasmic long dsRNA are particularly relevant [118].

PKR is ubiquitously expressed in mitochondria as well as in the cytoplasm in its unphosphorylated inactive form; its physiological role extends beyond an antiviral response [31,120]. PKR activation is strictly regulated during mitosis, and its activity is essential for proper cell division. The disruption of the nuclear structure during mitosis means that IRAlus escape compartmentalization and activate PKR. As a consequence, eukaryotic initiation factor 2α (eIF2α) becomes phosphorylated, with subsequent suppression of the global translation [121]. Inhibition of PKR by RNA interference or expression of a transdominant-negative mutant alleviating translation suppression during M phase lead to the dysregulation of several mitotic factors (cyclins A and B and polo-like kinase 1). The reduced phosphorylation of histone 3 and stabilization of G2-specific cell cycle regulators cause a delay in the progression from G2 to M phase [121]. Activated PKR also induces phosphorylation of p53, a tumor-suppressor protein with a pivotal role in controlling cell cycle and apoptosis, which leads to a 25–35% increase in cells arrested in G1. On the other hand, a reduction in PKR expression by doxorubicin decreases p53 stability [122,123].

Wound-induced hair neogenesis (WIHN) is a rare example of adult organogenesis in which dsRNA plays a central role [124]. The activation of TLR3 by endogenous dsRNA contributes essentially to wound healing and hair regeneration. Full thickness wounds in mice result in the release of dsRNA from damaged skin that activates TLR3 and triggers downstream signaling via interleukin 6 and STAT3 (Signal transducer and activator of transcription 3), which promote hair neogenesis. Moreover, activated TLR3 induces intrinsic synthesis of retinoic acid (RA) that orchestrates skin appendages’ growth and regeneration [125,126]. Injection of poly(I:C), a dsRNA analogue, into mouse wounded skin results in a significant increase in new hair formation, while TLR3-deficient mice failed to generate new hair upon skin wounding [124,126]. Furthermore, human skin biopsies taken after rejuvenation laser treatment display increased endogenous RA synthesis and enhanced gene expression signatures for dsRNA and RA [125].

Endogenous dsRNA and autoimmune diseases: Autoimmune diseases are pathologies where the immune system mistakenly attacks healthy cells. Around 50% of autoimmune diseases are of unknown etiology, while others are attributed to genetic pre-disposition or hormonal and environmental factors [127]. The contribution of dsRNA to autoimmune diseases was inferred by Schur and colleagues, who detected antibodies against dsRNA in the sera of 51% of patients with systemic lupus erythematosus (SLE) and 9% with rheumatoid arthritis as compared to 6% of normal people [128]. Elevated interferon levels and enhanced expression of IFN-stimulated genes in the blood of SLE patients have been shown more recently [129,130,131]. Furthermore, the presence of anti-MDA5 antibodies in dermatomyositis patients is considered as a prognostic marker associated with high death rate due to interstitial lung disease [132].

Myasthenia gravis is an autoimmune disease characterized by auto-antibodies against the acetyl choline receptor AChR. Injection of poly (I:C) in mice stimulates the expression of αAChR via TLR3 and PKR activation. Accordingly, the expressions of TLR3, PKR, IFR7, IRF5 and IFN-β are all upregulated in the thymus of patients with myasthenia gravis, indicating an important role of dsRNA signaling in the disease etiology [133]. PKR, MDA5 and RIG1 expression are all increased in psoriatic lesional skin, paralleled by high IFNα levels [134]. IFNα treatment for hepatitis C virus infection is well known to trigger autoimmune diseases such as psoriasis, antiphospholipid syndrome or sarcoidosis, highlighting the contribution of innate immunity to the pathogenesis of these diseases [135,136].

A-to-I RNA editing enhances transcriptome and protein diversity; conversely, editing in protein-coding regions generates auto-antigens and potentially causes or aggravates autoimmune diseases. Accordingly, increased editing was observed in SLE and rheumatoid arthritis [137,138]. On the other hand, there is a global reduction in A-to-I editing in psoriatic lesional skin and an accumulation of dsRNA feeding into an antiviral response, highlighting the fine balance between protective and detrimental consequences of dsRNA signaling.

dsRNA in cancer: Somatic mutations and escaping immune surveillance are essential steps in tumor initiation and progression. Recent studies have highlighted that RNA mutations constitute an additional cause for transition to malignant tumor, with RNA editing being a major cause for the underlying sequence changes. Adenosine-to-inosine changes in dsRNA by ADAR can give rise to transcriptomic alterations via point mutations, alternative splicing, altered RNA targeting and defects in microRNA synthesis [139]. Accordingly, many cancer types such as liver and breast cancer as well as some gastrointestinal malignancies express high levels of ADAR, which also promotes cancer growth and metastasis [140].

Although both ADAR1 and ADAR2 are linked to tumorigenesis, ADAR1 appears to play the major role due to its ubiquitous expression [139]. ADAR1 expression is stimulated by interferon as a negative feedback to control inflammation and cell survival, potentially also promoting tumor growth and invasiveness [141,142]. ADAR1 has been found to edit disease-relevant transcripts in a number of cancers [143]. For example, in prostate cancer, A-to-I editing in the androgen receptor transcript affects interaction of the receptor with androgens and androgen antagonists, which results in the reactivation of androgen signaling, tumor development and growth [144]. In hepatocellular carcinoma, increased levels of ADAR lead to editing of Antizyme Inhibitor 1 (AZIN1) and consequently enhanced nuclear import of the edited protein and stabilized interaction with its binding partner (Antizyme). The reduced inhibitory potential of the complex promotes tumor formation and is associated with aggressive behavior [145] (for a comprehensive review, see [143]).

dsRNA cancer therapies: There is a relation between autoimmune diseases and cancer—for example, long-standing autoimmune diseases may results in cancer transformation. Interestingly, the upregulation of ERV transcription is a common feature between these two pathologies [146,147]. The majority of ERVs are transcriptionally inactive, though 7% of their sequences can be reactivated by exogenous viruses or hypoxia [148]. Unlike in the autoimmune diseases discussed above, cancer cells mitigate the impact of moderate levels of ERV-related dsRNA formation and escape immune surveillance. However, drug-induced stimulation of ERV transcription can trigger a dsRNA-mediated immune response and make the cancerous cells visible to a variety of immune cells [149]. Hence, host dsRNA-binding proteins and the associated signaling cascades are widely used drug targets [150].

Transcription of ERVs is efficiently silenced through DNA hypermethylation in normal somatic cells [48]. Hypomethylating drugs such as azacytidine or decitabine induce transcription of ERVs and the formation of dsRNA, which, in turn, activates innate immune signaling. Both drugs are widely used to treat hematological cancers and have been investigated to treat other types of solid tissue tumors [151]. The consequences of bi-directional transcription of ERVs have been established in various cancer cell lines including epithelial ovarian cancers, colonic cancer cell lines and melanoma [48,152]. Accordingly, azacytidine causes an interferon response and increased expression of programmed death-ligand 1 (PD-L1), an important target in cancer immunotherapy [152].

A novel approach to treat various cancers combines ERV re-activation using histone deacetylase inhibitors (HDACi) in combination with immune checkpoint inhibitors targeting PD-1 or Cytotoxic T-Lymphocyte Associated Protein 4 (CTLA-4) [153]. Accordingly, ERV activation triggers a dsRNA-mediated interferon response that leads to increased expression of Major Histocompatibility Complex type I (MHC-I) on cancer cells; hence, the cell becomes “visible” to a T cell-mediated response [48]. Immune checkpoint inhibitors such as Atezolizumab and Avelumab or Ipilimumab (monoclonal antibodies against PD-1 or CTLA-4, respectively) used in combination dampen the inhibitory immune response and enhance anti-tumor activity [154].

The viral dsRNA analogues poly(I:C) and poly(A:U) are being used as adjuvants in anti-tumor therapy for their potential to stimulate an interferon response. There are two main mechanisms by which cancer cells are affected: first, by inducing cancer cell apoptosis through an IFN-β autocrine loop, and second, by IFN-β-mediated signaling. This leads to stimulation of the major players in anti-cancer immunity, including maturation and differentiation of dendritic cells, promotion of a T cell response and activation of natural killer cells [155]. Hence, immune-stimulatory adjuvants are key components of cancer vaccines together with tumor-specific antigens [156].

## 5. Conclusions

The pathways by which viral dsRNA activates innate immunity have been established for quite some time. In this context, the discovery of widespread dsRNA formation from endogenous sources such as repetitive elements or natural antisense transcripts raised questions of how the different stimulators of innate immunity are controlled. Compartmentalization and specialized dsRNA sensor proteins that integrate structural information and dsRNA abundance to elicit a physiologically sensible response have evolved as a protective strategy. Nonetheless, cellular dsRNA homeostasis is often challenged in disease and these observations have disclosed an interplay between repetitive genomic elements, long non-coding RNA and innate immune signaling that can jeopardize the well-being of cells, organs and the entire organism. A detailed understanding of dsRNA expression and processing can inform strategies to avoid ectopic dsRNA formation and inflammation through stress, drugs or malnutrition, for example. Alternatively, therapeutic stimulation of dsRNA expression shows great promise in directing an immune response against cancer cells.

## Figures and Tables

**Figure 1 ncrna-07-00015-f001:**
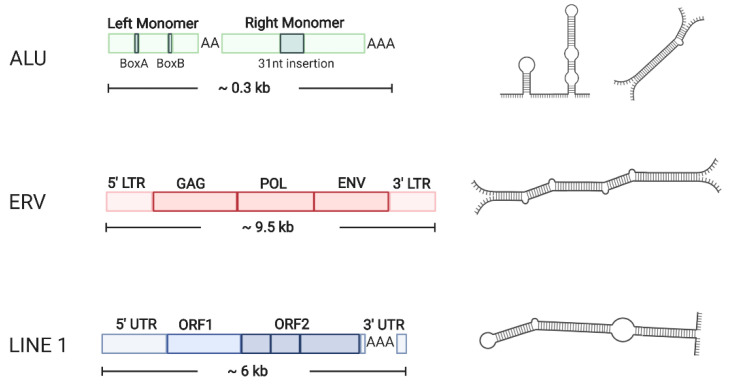
Schematic representation of repetitive elements in the human genome associated with double-stranded RNA (dsRNA) formation. LINE 1 and endogenous retroviruses (ERVs) give potential rise to long dsRNA structures formed from convergent transcripts or hairpin structures from read-through transcription of head-to-head/tail-to-tail arranged elements. Alu elements are much shorter and form hairpin structures as well as “open” dsRNA hybrids, though the intermolecular duplexes are rare. Alu elements are the predominant target for adenosine deaminase acting on RNA (ADAR)-mediated adenosine-to-inosine (A-to-I) editing. LTRs function as bi-directional promoters. ORF, open reading frame; GAG (group specific antigen), POL (reverse transcriptase), ENV (envelope protein), retroviral proteins; UTR, untranslated region; LTR, long terminal repeat. Figure created with Biorender.com.

**Figure 2 ncrna-07-00015-f002:**
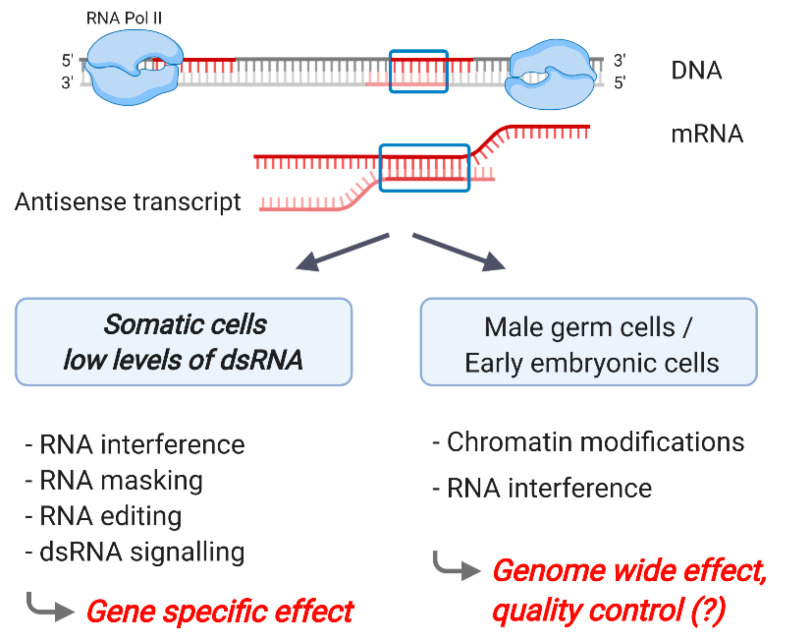
Double-stranded RNA (dsRNA) formation from sense–antisense transcripts. Natural antisense transcripts are processed and potentially reach the cytoplasm, where they interact with the sense transcript. In somatic cells, the level of sense–antisense hybrids is low, and there is no evidence of ADAR editing, for example, nor is dsRNA immune signaling triggered. Various mechanisms (RNA interference, RNA masking, RNA editing and dsRNA signaling) are potentially triggered by the dsRNA, depending on the cellular context. In male germ cells and during early embryogenesis, sense–antisense dsRNA formation may play a general, system-relevant role. Figure created with Biorender.com.

**Figure 3 ncrna-07-00015-f003:**
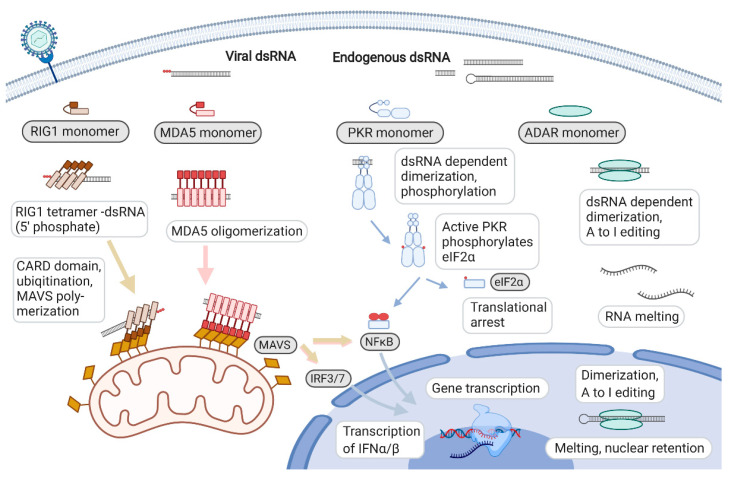
Double-stranded RNA (dsRNA) sensor proteins and activation of innate immunity. Viral dsRNA (including 5′ phosphorylation) or dsRNA from mitochondria and repetitive elements in the cytoplasm are recognized by dsRNA sensors retinoic acid-inducible gene I (RIG1), melanoma differentiation-associated gene 5 (MDA5), protein kinase R (PKR) and ADAR. RIG1 requires the 5′ phosphate group to initiate oligomerization, and MDA5 forms long dsRNA-dependent polymers. Both structures induce mitochondrial antiviral signaling (MAVS) polymerization and, eventually, caspase and interferon signaling. PKR binds short dsRNA molecules, dimerizes and becomes activated through autophosphorylation. Activated PKR dissociates from dsRNA, phosphorylates eukaryotic initiation factor 2α (eIF2α) (which, in turn, inhibits translation globally) and triggers an interferon response. ADAR is present in both the nucleus and cytoplasm and antagonizes dsRNA signaling by melting the RNA hybrid. Figure created with Biorender.com.

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
