# Peer review of "Endogenous Double-Stranded RNA"

_ncrna, 2021, doi:10.3390/ncrna7010015_

Round 1
Reviewer 1 Report
The review about dsRNAs by Sadeq et al. is well balanced and will be of great interest to the readership of the journal
My main points to add:
- please elaborate more on the somatic vs. germline/ESC phenotype (Figure 2). Explain briefly the role of dsRNAs in the female germline. The male germline is an immune-privileged and thus, will not promote the immune response seen in the somatic tissue, which is certainly a point to discuss.
- Add to the conclusion where you see that the field will be developing or should pay more attention to.
I suggest minor amendments:
- title: The authors do not refer to the "CO2 in the world of lncRNAs" in the manuscript. I would refrain from using this part as some readers might not get interested in reading the review because of the assumption that the authors describe dsRNA-related phenomena in plants (which in itself is another large review and beyond the scope of this review).
- abbreviations: please describe the abbreviations of genes/proteins when mentioned first (e.g. MDA5 and LGP2, line 341).
- abstract: remove MDA5 and PKR, line 21
- explain briefly MDA5 and RIG-1, line 176. Perhaps the authors can move the description from line 194
- please refer in the text to the figures
- please correct some problems with the citations, e.g., line 309 citation 74
- a citations are missing, e.g., Werner et al, under revision. I assume this manuscript is from the same group. Can the authors deposit the article to bioRxiv and cite it?
- Does Biorender allow publishing of the illustrations?
Author Response
I would like to thank the reviewer for the valuable comments. I have amended the manuscript accordingly, all changes are highlighted in the file 'R1-MarkUp. Concerning the suggestion to submit my paper 'under revision' to BioRxiv, I plan to resubmit the final version by the end of this week and would rather spend my time on the revision than on a preprint submission. I have contacted the editors and they have approved my strategy. Moreover, I have taken a subscription for 'BioRender' which allows the export of high resolution figures and publication.
With kind regards
Andi Werner
Reviewer 2 Report
The paper titled " Double-stranded RNA, the CO2 in the world of lncRNAs” provides a broad picture of cellular dsRNA sources and how do the cellular proteins respond to dsRNAs. In addition, this review also described the role of dsRNAs in cancer and auto immune diseases and how could these dsRNAs can be used as potential therapeutics to cancers. The review is well written, comprehensive and informative. I find the review valuable to the field.
Here are my comments to authors.
Major comments:
- Page2-line 91. I found this discussion on how mtRNAs serves as one of the main sources of cellular dsRNAs is comprehensive. In the last paragraph of this section, authors make a comment on how stress mediates mtRNA leakage to cytoplasm. I think it’s worth mentioning that the defective mitochondrial RNA turnover (by SUV3, PNPase, or MTPAP) also leads to the accumulation mtRNAs in the cytoplasm of cells, which is associated with an altered immune response as these proteins (SUV3, PNPase, or MTPAP) are being discussed in the previous paragraph ((Pajak et al. 2019) PLoS Genet 15: e1008240).
- Page 3, line 139- “The repetitive nature of Alu insertions allows the formation of both inter- and intra-molecular dsRNA which is recognized by the nuclear isoform of ADAR (adenosine de-aminase acting on RNA)”. Reference is missing for this fact. In addition, this sentence needs to be re-phrased; Inversely oriented Alu repeats generally form intra-molecular duplexes, in contrast intermolecular duplexes are rarely formed ((Kawahara and Nishikura 2006) FEBS Lett 580: 2301-2305).
- Page 8, line 329- The reference cited for describing the interaction between dsRNA binding proteins (dsRBPs) and RNA is a review article that discuss the origins of specificity in Protein-DNA recognition. A handful of review articles are available for dsRNA recognition by dsRBPs (for example: (Masliah et al. 2013) Cell Mol Life Sci 70: 1875-1895). In addition, sequence specific readouts of minor grooves are reported in ADAR2 double stranded RNA binding domain (dsRBD) in complex with an RNA stem–loop deriving from a natural substrate of ADAR2 ((Stefl et al. 2010) Cell 143: 225-237).
- Page 10, line 414- This sentence needs to be re-phrased; “ADAR2 is specific for certain transcripts expressed in the central nervous system and acts on small hairpins in the coding region of neurotransmitter receptors”.
5. ADAR2 is expressed in most mammalian tissue with brain and lung expressing the highest levels ((Melcher et al. 1996) Nature 379: 460-464). In addition, there are instances where ADAR2 edits within the coding region of proteins outside of nervous system ((Stulic and Jantsch 2013) RNA Biol 10: 1611-1617).
Minor comments:
- ‘Adenosine to Inosine editing’ is introduced in page 2, line 80. Use the ‘A to I editing’ here after. Both ‘Adenosine to Inosine’ and ‘A to I’ are used throughout this review article. Introduce a new term with abbreviation for the first time of use and continue with the abbreviation throughout the review. This is true for ‘Mitochondrial antiviral signaling’ and ‘MAVS’ , ‘untranslated regions’ and ‘UTR’ as well.
- Page 3, line 108- typo in PKR fCLIP-seq
- In Figure1, change the order of LINE 1, ERV and ALU in the order of discussion in the text.
- Page 7, line 309- reference formatting.
- Page 12, line 508- ‘In various cancers’ needs to be replaced with hepatocellular carcinoma as the rest of the sentence specifically describes a specific editing event.
Author Response
I would like to thank the reviewer for the valuable comments. I have amended the manuscript accordingly, all changes are highlighted in the file 'R1-MarkUp. In particular, I have clarified the paragraphs concerning dsRNA formation among Alu repeats and ADAR action in brain and other tissues. The citation that the reviewer kindly brought to my attention are also included in the updated bibliography.
Many thanks and kind regards
Andi Werner